# Effectiveness of the Combined Use of a Brain–Machine Interface System and Virtual Reality as a Therapeutic Approach in Patients with Spinal Cord Injury: A Systematic Review

**DOI:** 10.3390/healthcare11243189

**Published:** 2023-12-17

**Authors:** Amaranta De Miguel-Rubio, Ignacio Gallego-Aguayo, Maria Dolores De Miguel-Rubio, Mariana Arias-Avila, David Lucena-Anton, Alvaro Alba-Rueda

**Affiliations:** 1Department of Nursing, Pharmacology and Physiotherapy, University of Cordoba, 14004 Cordoba, Spain; n92gaagi@uco.es (I.G.-A.); b42alrua@uco.es (A.A.-R.); 2Ilerna Private Teaching Center, 14011 Cordoba, Spain; lolademiguel@ilerna.com; 3Physical Therapy Department, Universidade Federal de São Carlos, São Paulo 13565-905, Brazil; m.avila@ufscar.br; 4Department of Nursing and Physiotherapy, University of Cadiz, 11009 Cadiz, Spain; david.lucena@uca.es

**Keywords:** spinal cord injuries, virtual reality, brain–computer interfaces, rehabilitation

## Abstract

Spinal cord injury has a major impact on both the individual and society. This damage can cause permanent loss of sensorimotor functions, leading to structural and functional changes in somatotopic regions of the spinal cord. The combined use of a brain–machine interface and virtual reality offers a therapeutic alternative to be considered in the treatment of this pathology. This systematic review aimed to evaluate the effectiveness of the combined use of virtual reality and the brain–machine interface in the treatment of spinal cord injuries. A search was performed in PubMed, Web of Science, PEDro, Cochrane Central Register of Controlled Trials, CINAHL, Scopus, and Medline, including articles published from the beginning of each database until January 2023. Articles were selected based on strict inclusion and exclusion criteria. The Cochrane Collaboration’s tool was used to assess the risk of bias and the PEDro scale and SCIRE systems were used to evaluate the methodological quality of the studies. Eleven articles were selected from a total of eighty-two. Statistically significant changes were found in the upper limb, involving improvements in shoulder and upper arm mobility, and weaker muscles were strengthened. In conclusion, most of the articles analyzed used the electroencephalogram as a measurement instrument for the assessment of various parameters, and most studies have shown improvements. Nonetheless, further research is needed with a larger sample size and long-term follow-up to establish conclusive results regarding the effect size of these interventions.

## 1. Introduction

Spinal cord injury (SCI) has a major impact on both the individual and society. Hence, an increasing number of professionals are involved in the treatment of affected individuals, seeking the most advanced techniques for the enhancement of patient recovery [1]. Considering that a SCI entails a chronic life situation, the impact on the health system is not limited to the acute phase of the injury, rather, the person with SCI must face chronic diseases derived from the injury during their entire lifetime [2]. Spinal cord damage can cause a permanent loss of sensorimotor functions and persistent neuropathic pain, leading to structural and functional changes in the spinal cord [3]. In addition, most patients experience difficulties performing certain activities of daily living, which may lead to a poorer perception of quality of life [4,5].

Complete or incomplete SCI causes persistent neurological deficits because of the interruption of nerve impulses. The creation of a glial scar from the continuous deposition of fibrous tissue generates a physical barrier for axonal regeneration because of the main damage resulting from the injury [6,7]. The neurological severity of a SCI is commonly graded according to the American Spinal Injury Association Impairment Scale (AIS). This scale assesses motor and sensory functions and groups patients with SCI into five functional categories from A (absence of both functions) to E (normal function or with minimal neurological deficit) [8].

A brain–machine interface (BMI) can help restore independence to people with paralysis by using brain signals to control prostheses or trigger functional electrical stimulation [9]. The possibility of establishing a direct channel of communication and control between the human brain and computers or robots has been the subject of scientific speculation and even science fiction for many years [10]. This technology, called brain–machine interface (BMI) technology, provides a new output channel for brain signals to communicate with, or control external devices without using the normal output pathways of peripheral nerves and muscles. A BMI recognizes the user’s intent through electrophysiological or other brain signals. Electrophysiological signals may be recorded on the scalp, under the scalp, or within the brain; other types of physiological signals may be recorded by magnetic sensors or other means [11]. Other methods of assessment based on clinical neurophysiology, such as evoked potentials or event-related potential have demonstrated sensitivity for a range of cognitive functions, including attention, language processing, and memory [12]. In real time, a brain signal is translated into output commands that fulfill the user’s wish. The most common example of the use of such technology is the direct control of a computer cursor by a BMI based on electrophysiological signals [13]. The development of BMI systems has largely focused on improving the functional independence of people with severe motor disabilities, including the provision of tools for communication and mobility [14].

Virtual reality (VR) is another revolutionary therapy that can help address certain impairments caused by a SCI. Thus, VR can offer patients novel challenges and difficulties, offering a “training” that may enable people to learn possible responses that can be applied in their daily lives [15]. The use of virtual reality training can play an important role in improving cognitive functions and motor disabilities [16,17,18]. Virtual environments are offered with different degrees of immersion: non-immersive, partial, and total [19,20,21,22,23]. Total immersive VR has been gaining attention following explosive growth in VR technologies over the past decade. The key to such success is attributed to the realistic immersive settings that the head-mounted displays can produce and provide users [24]. These screens display the scene in first person, and each eye is shown slightly different two-dimensional images, thus creating the illusion that the person is seeing a three-dimensional environment [25].

Neuroprostheses that combine a BMI with functional electrical stimulation (FES) can restore voluntary control of patients’ paralyzed limbs [26]. After years of research, there is evidence that subjects can improve with VR using an appropriate BMI that is able to adapt to the patient and which, in turn, the patient is able to adapt to; however, this therapy has focused mainly on people with cerebral palsy and stroke [27,28]. Thanks to the latest technological advances, this treatment can be performed from home, which means that patients will be able to improve more rapidly with the help of their physiotherapists, occupational therapists, and family members [29]. Therefore, the aim of this systematic review was to evaluate the effectiveness of the combined use of VR and BMI in patients with SCI.

## 2. Materials and Methods

### 2.1. Literature Search

The PRISMA guidelines (Preferred Reporting Items for Systematic Reviews and Meta-Analyses) were followed in this SR [30]. Appendix A features a complete PRISMA checklist. The search protocol was registered in the International Prospective Register of Systematic Reviews (PROSPERO) database (CRD42023352246). 

The literature search was conducted in the following databases: PubMed, Web of Science, PEDro, Cochrane Central Register of Controlled Trials, CINAHL, Scopus, and Medline, including articles published from the beginning of each database until January 2023. The search strategy was as follows: (“spinal cord injury” OR “spinal cord injuries” [MeSH] OR “paraplegia” [MeSH] OR “tetraplegia” OR “wheelchair”) AND (“virtual reality” [MeSH] OR “virtual reality exposure therapy” [MeSH] OR “virtual systems” OR “augmented reality” [MeSH] OR “virtual environments” OR “video games” [MeSH] OR “exergames”) AND (“brain computer interfaces” [MeSH] OR “body machine interface”). The PubMed search was performed using the MeSH descriptors (Appendix A). Hand searches were also performed, by searching the reference list of studies included in the review, and adding those studies that met the inclusion criteria. 

### 2.2. Selection Criteria

The PICOS model was used [31] (Population, Intervention, Comparison, Outcome/Outcome, Study design), to define the inclusion criteria: (P) Population; adults diagnosed with SCI. (I) Intervention; immersive, semi-immersive, or non-immersive VR combined with a BMI system connected to SCI patients. (C) Comparison; other intervention: no treatment, usual care/activities, conventional rehabilitation program, traditional physical, or occupational therapy treatment. (O) Outcome; proprioception. level of pain. Kinesthetic motor imagery. Upper extremity muscle strength. General shoulder function. Range of motion of the upper limb joints. (S) Study design; no restrictions related to study design.

Exclusion criteria; studies involving other pathologies in addition to SCI without providing separate details of the results between populations. Publications in the form of summaries and reviews.

Two reviewers (A.D.M.-R. and I.G.-A.) independently screened, reviewed, and extracted data from the final studies. In the event of any doubts or discrepancies, a third reviewer (A.A.-R.) participated in this process. 

### 2.3. Data Extraction 

The information extracted from each article included: author, country, number of participants, age and sex, AIS grade, level of injury, time since injury, type of study, level of evidence, type of intervention, session intensity, session duration, intervention duration, study variables, measurement instruments, and results. 

### 2.4. Quality Assessment 

Risk of bias was evaluated using the Cochrane Collaboration’s tool [32], developed by the Review Manager 5.3 software (Copenhagen, Denmark). This tool provides an evaluation of different items according to risk of bias. Studies are categorized as: “unclear risk”, “low risk”, and “high risk”. The risk of bias assessment was conducted by two reviewers. When in doubt, a third assessor was involved in the final decision. In order to evaluate the methodological quality of the studies, the Spinal Cord Injury Rehabilitation Evidence (SCIRE) system [33] and the Physiotherapy Evidence Database (PEDro) scale [34] were used. Moreover, the level of evidence of the included studies was classified using the combination of the SCIRE and PEDro systems. This combined score (SCIRE-PEDro) uses different categories to analyze the research design and methodological quality, grading from level 1 (highest quality) to 5 (lowest quality). The methodological quality of the studies was assessed using the PEDro scale. This scale features items related to selection, performance, detection, information, and attribution bases. According to the PEDro scale, research with a score of 9–10 is considered methodologically excellent, while a score of 6–8 is good. 

### 2.5. Data Synthesis

A systematic review was conducted using qualitative synthesis, considering the heterogeneity of the variables studied and the treatments included in the trials. For this reason, a meta-analysis (quantitative synthesis) could not be performed. 

## 3. Results

The literature search yielded a total of 82 articles from the electronic databases, of which 31 were duplicates. Of the 51 remaining articles, those that were unrelated to the study aim were removed (19), which resulted in a total of 32 articles. Those studies that did not combine VR with BMI, treated other pathologies, or were incomplete were removed, leaving 10 articles, which, together with those found by other means (1), yielded 11 final papers. The flow chart for the selection of the articles included in this SR was based on the PRISMA recommendations [30], displayed in Figure 1.

The 11 selected articles were those by: Abdollahi et al. [35], Bayon-Calatayud et al. [36], Casadio et al. [37], King et al. [38], Leeb et al. [39], Mason et al. [40], Nicolelis et al. [41], Pais-Vieira et al. [42], Salisbury et al. [43], Tidoni et al. [11], and Wang et al. [44]. Table 1 and Table 2 show the main characteristics analyzed in the 11 selected articles.

### 3.1. Summary of the Main Results

Nine articles of the eleven selected used electroencephalogram (EEG) as a measurement tool, with the exception of Abdollahi et al. [35] and Casadio et al. [37], who used the BoMI Controller and MMT. In addition, all articles except Mason et al. [40] used the AIS scale to clarify the level of SCI; however, this article does not provide the necessary information to classify its participants on this scale. The most relevant results were greater precision in the movements requested [35,39,40], improved grip in the affected arm [36,40], improved online (VR) performance of participants [38], progress was made in terms of the initial classification of SCI, which evolved from AIS A to AIS C scale [41] improved performance during the sessions [42], viability [43], the patients’ sense of embodiment within the VR [11], and a realistic approach to the treatment of patients with SCI was appreciated [44].

### 3.2. Assessment of the Risk of Bias and Methodological Quality of the Studies Included in the Review

Figure 2 and Figure 3 show a summary of the risk of bias assessment of the included studies, both globally and individually for each study. When analyzed individually (Figure 2), the study by Nicolelis et al. [41] has the lowest risk of bias; conversely, the studies with the highest risk of bias are those by Bayon-Calatayud et al. [36], Leeb et al. [39], and Pais-Vieira et al. [42]. Overall (Figure 3), 100% of the biases appear when assessing performance biases. Furthermore, regarding the risk of bias among the analyzed studies, the lowest biases were found with the selective reporting of results (0%) and partial reporting (18%), while the highest value (100%) was found for allocation concealment. 

The methodological quality of the only RCT found in this SR was good (total PEDro score = 7) (Appendix A). The remaining studies obtained a level four and five level of evidence according to the SCIRE-PEDro criteria (Table 2).

## 4. Discussion

The aim of the present SR was to estimate the feasibility of treatment combining VR with BCI in patients with SCI. Eleven articles were selected for this study, of which only one was a randomized controlled trial (RCT) [41], three were case studies [36,39,42], two were post-test [11,44], and five were pre-post-test [35,37,38,40,43] studies. All articles share a series of common characteristics useful for the present study, which, when compared, help us to answer our question. Of the total sample (*n* = 93), the number of participants per study were between a minimum of five and a maximum of twenty-five, without counting the articles that only analyzed one case, such as Bayon-Calatayud et al., Leeb et al., and Pais-Vieira et al. [36,39,42]. The participants included 75 men and 18 women. The mean age was above 18 years in all cases, with the lowest mean age belonging to the intervention groups (IG) in the studies of Tidoni et al. [11] and Wang et al. [44]. Participants in most papers were between 21 and 64 years of age, except for the study by Nicolelis et al. [41], which only reports that the participants are over 18 years of age. A total of fifty-one patients presented a complete SCI, four were incomplete, and one was not described. Of the complete lesions, 62.7% were cervical (32 lesions), 27.4% were thoracic, and 1.9% were lumbar. There were 34 healthy patients among all the articles, representing individuals who were part of the control group (CG), which helped to validate the results.

In all the studies, the CG and IG underwent the same treatment, and therefore the results validate the true effect in individuals with SCI compared to healthy individuals. The total number of sessions received ranged from 5 to 28, divided between 10 days and 4 months, although some authors fail to specify this information, concretely: Salisbury et al. [43], Tidoni et al. [11], and Wang et al. [44].

VR and BMI have been supported by other innovative techniques that provide a more realistic and differentiated view of the treatment given to patients today, such as BoMI, a customized cervical LM BMI system [35], treatment using BMI with VR, and electro-functional electrostimulation (BCI + FES + VR) combined with occupational therapy [36], LF-ASD, an EEG-based brain switch that allows the patient to turn a video game character in real time by thinking that character is moving in that direction [40], a treatment that integrates assisted locomotion with noninvasive BMI, VR, and tactile feedback [41], a protocol comprising VR goggles, tactile and thermal feedback sleeves, headsets and controllers to provide a much more immersive experience [42], and finally, treatment using kinesthetic motor imagery (KMI) to move an avatar forward in a VR environment, and inactivity to stop [44]. 

BMI systems are used for severe motor restrictions, combined with the use of external movement aids, although they can also be used for basic rehabilitation purposes [45]. Thus, BMI provides us with a new tool to restore mobility in paralyzed limbs [46]. In BMI, most closed-loop stimulation applications act on peripheral nerves or muscles, resulting in rapid muscle fatigue [47].

The AIS scale has been chosen by 10 of the 11 articles to measure the degree of SCI, with the exception of Mason et al. [40], who does not mention this scale at any time, nor does it label its participants in any of its grades. Another key measurement tool of the selected articles is the EEG, which appears in 9 of the 11 articles, Abdollahi et al. [35] and Casadio et al. [37] are the only two that do not use it. The EEG can be helpful for perceiving the nervous response that the patient is going to have in the area of interest while immersed in VR by means of the BMI [48].

An important consideration in the selected articles is the measurement of parameters using the upper extremities as a reference point, since in Abdollahi et al. [35], Bayon-Calatayud et al. [36], Casadio et al. [37], Leeb et al. [39], Pais-Vieira et al. [42], and Tidoni et al. [11], a sensor is placed on the hand, shoulder, biceps, or even the sleeve to capture information about the movements and their location.

Abdollahi et al. [35] uses inertial measurement units (inserted into a custom-made vest) to capture localized body movements, allowing the patients in this study (with cervical injury) much greater accuracy in detecting their movements, which will help them through practice to be more precise in their tasks. The BoMI used in the study can simulate motor learning and potentially overcome established barriers for independence and partial recovery in patients with cervical SCI. Treatment with BoMI offers patients the possibility of controlling their chairs by means of residual movements of the upper third of the body. Thus, although the process followed by Casadio et al. [37] for obtaining responses is somewhat different, using four infrared video cameras (two-dimensional each) to track four active light markers, both studies analyze the residual movements that SCI patients have, using them to their advantage by assigning meaning to them. 

One of the most innovative therapies of the selected articles is the combination of BMI, functional electrical stimulation, and virtual feedback proposed by Bayon-Calatayud et al. [36]. At the end of the treatment intervention, the patient completed a usability questionnaire to evaluate the feasibility of the project. The accuracy of the patients with this technique after finishing the five indicated sessions was 85.8 ± 11.8%. Both this study and those by Leeb et al. [39] and Pais-Vieira et al. [42] reveal favorable and promising results; however, for greater reliability, more studies are needed with larger sample numbers to demonstrate the results in a larger number of patients. 

Both Leeb et al. [39] and King et al. [38] used similar methods regarding the information given to the patient when involved in VR and BMI. In both studies, the patient should start by creating an idling and walking KMI; these data are then recorded with the EEG. Upon beginning data collection, Leeb et al. [39] use a form of immersion with three “cave” projections on the three walls surrounding the patient and a screen in the patient’s frontal field. This multi-wall projection system has a special feature in that the images on adjacent walls are seamlessly joined together without leaving sharp corners. Moreover, King et al. [38] uses a screen to project the image on a monitor. The high level of control achieved in both studies of SCI patients gives us some optimism for the development of lower limb prostheses controlled by an BMI system, and the protocols used for these studies can be used as tools for greater accuracy and control of BMI [38,39]. These authors are joined by Wang et al. [44] with their KMI system including idling and running moments. 

A relevant aspect of the study by Salisbury et al. [43] are the headsets used (Emotiv EPOC), initially geared towards video games, but also incorporated in numerous studies; these are compact wireless headsets that require minimal effort to set up and allow much more flexibility and mobility than traditional EEG, and even analyze the patient’s facial features during their required activity within the study. Both this article and those by Abdollahi et al. [35], Tidoni et al. [11], and Mason et al. [40] modify their treatment protocol and instead of performing processed gait in VR, they practice fine motor techniques. 

Pais-Vieira et al. [42] employed a BMI configuration for neurorehabilitation, combining EEG activity, VR (visual and auditory), and tactile and thermal feedback sleeves in patients with SCI to determine whether this combination of multimodal feedback would prevent brain control of an avatar. The patient was able to modulate neural activity to generate “Walk” and “Do not walk” commands according to the cues presented, supporting the hypothesis that this multimodal feedback did not impede the avatar’s brain control. An interesting finding of this study is that the patient reported feeling cold in the lower extremities when his avatar was placed in a water setting [42].

In the study by Nicolelis et al. [41], after following the action protocol prepared by the physicians, the participants managed to improve their status on the AIS scale, which is quite encouraging, since it means a significant improvement in terms of the patient’s neurological activity. 

### Limitations

It was difficult to find high-quality articles that combined VR treatment with a BMI system in patients with SCI in the same study. Moreover, most of the studies did not have a CG, which reduces the quality of the study, and they also included a rather small sample of patients, making it difficult to generalize the study to the entire population. The levels of SCI differ greatly from paper to paper, thus modifying the approach of the study methodology, which may be aimed at improving the patient’s gait or at improving their function by optimizing their voluntary movement of the upper extremities. Also, none of the studies provide information on the patient’s injury status months after the intervention, and therefore the long-term effect of the treatment is unknown, which may also be influenced by the novelty of this technique. Finally, it is important to consider that although an artificial intelligence tool may be useful for use in systematic reviews, it may also have inherent limitations regarding its ability to retrieve all works relating to the problem. This should be considered as a potential limitation and, therefore, human oversight is potentially necessary. 

## 5. Conclusions

In the studies analyzed in this SR, the combined treatment with VR and BMI can be carried out in two manners, depending on the purpose of therapy. 

A first aspect is more related to the recovery of the patient’s gait, which is the patient’s main concern, and for this purpose, a neurological response below the level of injury has been sought by means of BMI systems immersed in VR, which forces the patient to have intentionality of gait, and this reactivation can be favored. 

The other aspect of treatment focuses on wheelchair-bound patients who have low motor activity, even in the upper limbs, and who, through their residual upper trunk skills and a trained BMI in a VR environment, may have sufficient autonomy to not rely on a third person to carry out their daily functions.

Most of the articles analyzed have used the EEG as a measurement tool for the assessment of various parameters, using the upper limbs as a reference point. With all the systems used, improvements have been obtained in most of the parameters analyzed, although the statistically significant changes have occurred in the upper limb, where the mobility of the shoulder and upper arm has improved, and the weakest muscles have been strengthened.

The improvement of patients in terms of BMI connection over the course of the sessions is clear and encourages us to be very optimistic about this therapy, as good results have been obtained every time it has been used. However, it would be interesting for future research to group the different patients according to their degree of SCI to determine the most appropriate type of treatment, analyze protocols with larger samples, and to increase the number of intervention sessions.

## Figures and Tables

**Figure 1 healthcare-11-03189-f001:**
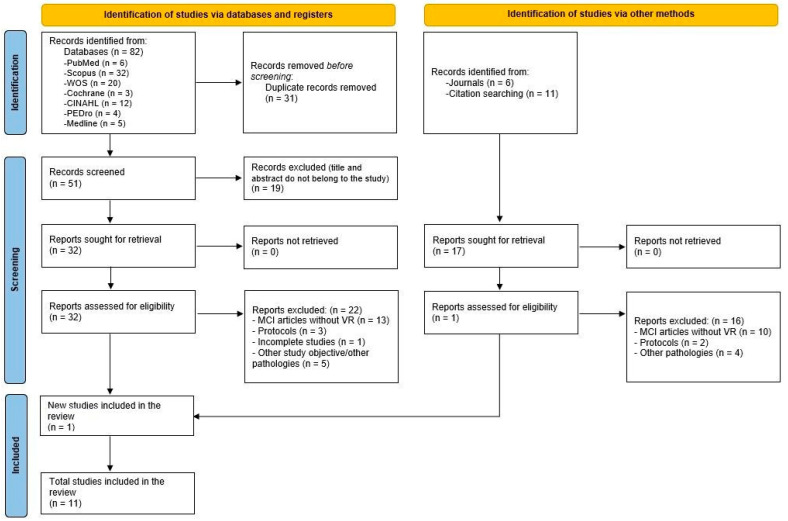
Flow diagram of the selection process of the systematic review following the PRISMA recommendations [30].

**Figure 2 healthcare-11-03189-f002:**
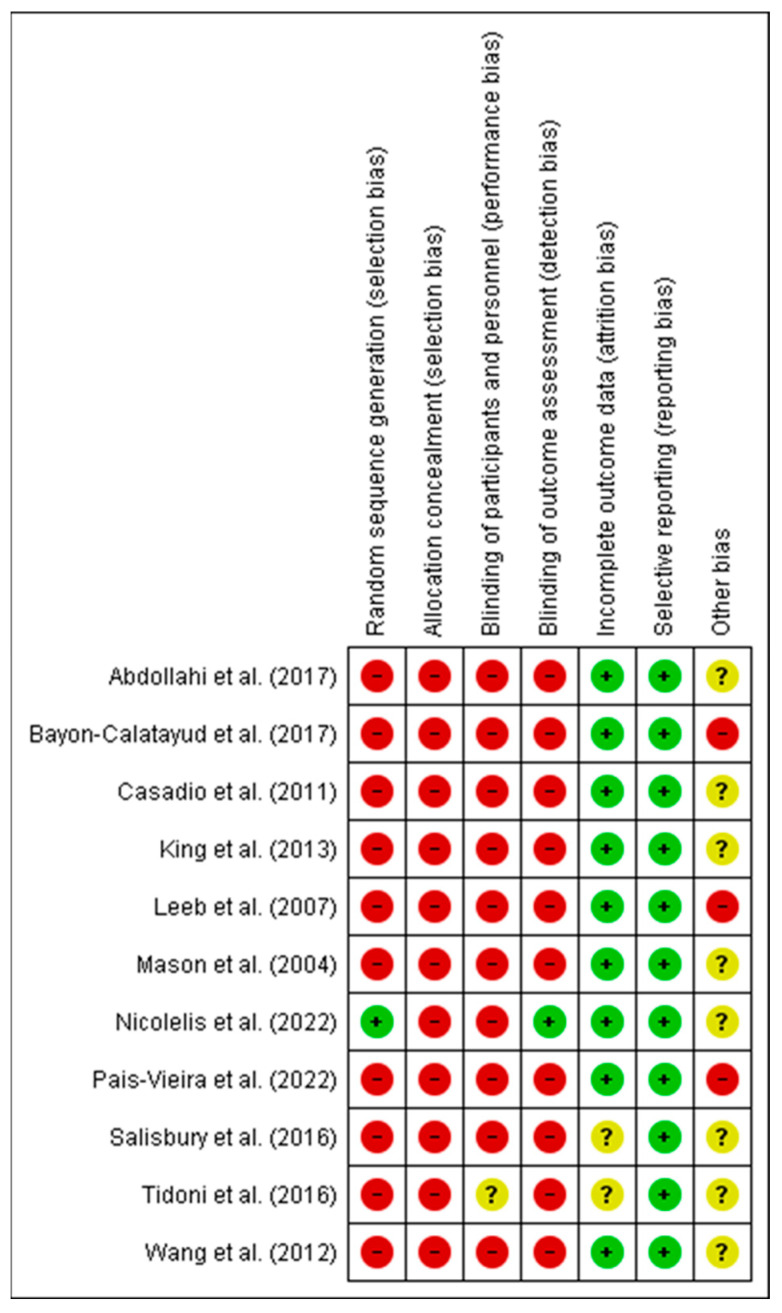
Risk of bias of the studies included in the systematic review. The green circle (+) indicates low risk of bias, the yellow circle (?) unclear risk of bias, and the red circle (-) high risk of bias [11,35,36,37,38,39,40,41,42,43,44].

**Figure 3 healthcare-11-03189-f003:**
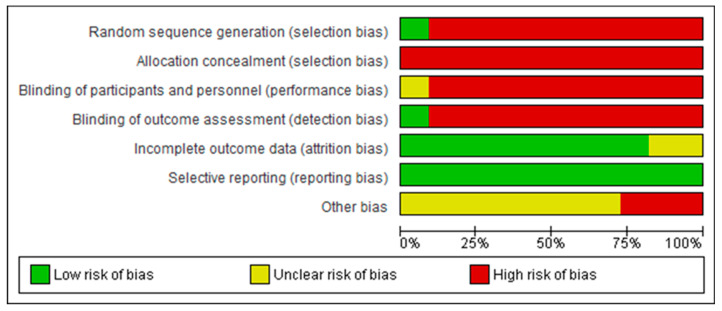
Overall risk of bias, with each category presented as percentages.

**Table 1 healthcare-11-03189-t001:** Demographic and clinical characteristics of the studies.

Country of Study	Participants (*n*)	Age (Years) Mean ± SD. (Ranged)	Sex M/F	AIS Grade	Level of Injury (Number of Participants with Each Level)	Time after Onset Injury (Months)
Abdollahi et al., 2017 [35] (Chicago) USA	*n* = 8	(29–58)	5M/3F	6A/2B	Complete cervical: C4 (2), C5 (2), C5-C6 (1), C6 (3)	150
Bayon-Calatayud et al., 2017 [36] (Toledo) Spain	*n* = 1	55	1M	D	C5	4
Casadio et al., 2011 [37] (Chicago) USA	*n* = 14 IG: 6 CG: 8	IG: 40.17 ± 3.53 (28–56) CG: (21–35)	IG: 6M CG: 7M/1F	IG: A (3) C (3) CG: ND	Complete cervical: C4 (1), C5 (1), C6 (1) Incomplete cervical: C3–C4 (1), C4 (2)	50.83
King et al., 2013 [38] (California) USA	*n* = 5	40.6 ± 18.4 (21–59)	5M	A-B	Complete cervical: C5 (1) Complete thoracic: T1 (2), T11 (2).	3.62 ± 2.12
Leeb et al., 2007 [39] (Graz) Austria	*n* = 1	33	1M	ND	Complete cervical: C5	108
Mason et al., 2004 [40] Canada	*n* = 8 IG: 4 CG: 4	IG: (33–56) CG: (31–56)	IG: 4M CG:3M/1F	ND	C4–C5 (3) C5–C6 (1)	147
Nicolelis et al., 2022 [41] (São Paulo) Brazil	*n* = 8 IG: 4 CG: 4	>18	IG: 4M CG:4M	A	Thoracic	21
Pais-Vieira et al., 2022 [42] (Oliveira) Portugal	*n* = 1	52	M	A	Thoracic: T4	240
Salisbury et al., 2016 [43] (Dallas) USA	*n* = 25	45 ± 13.0 (18–64)	19M/6F	A (4), B (7), C (7), D (5), ND (2)	Cervical: (12) Thoracic: (11) Lumbar: (1) ND: (1)	1.66
Tidoni et al., 2016 [11] (Rome) Italy	*n* = 13 IG: 3 CG: 10	IG: 28 ± 5.19 (22–31) CG: 29.33 ± 2.87 (24–32)	IG: 3M CG: 4F/6M	A–B–D	Complete cervical: C4 (1), C4-C5 (1) Incomplete cervical: C6 (1)	88.67
Wang et al., 2012 [44] (California) USA	*n* = 9 IG: 1 CG: 8	IG: 27 GC: (21–57)	IG:1F CG: 6M/3F	B	Thoracic: T8	132

AIS: American Spinal Injury Association Impairment Scale; F: female; CG: control group; IG: intervention group; M: male; ND: not described.

**Table 2 healthcare-11-03189-t002:** Main characteristics of the interventions.

Study SCIRE-PEDro Scores	Group Interventions	Intensity	Session Duration	Intervention Duration	Outcome	Measurement Instrument	Results
Abdollahi et al., 2017 [35] Pre-post test Level 4	Patients with SCI wore a garment with sensors on their shoulders to perform three actions in each session: reaching, writing, and playing.	2×/ week	1 h	12 weeks	Movements and time to perform them, movement error, ball striking, writing, standardized pull, reaching routes.	AIS, Cursor/motion sensor, stopwatch, BoMI controller.	Motion accuracy: 1st session 7.12 to last 2.85 s. (*p* < 0.001). Straighter reaching paths from 0.71 to 0.24 (*p* < 0.001). Pong hit rate from 5.26 to 19.59 min^−1^ (*p* < 0.001). Typing rate from 8.56 to 14.67 characters/min (*p* < 0.006). Movement error from 6.09 to 1.75 cm (*p* < 0.001). Movement jerks (smoothness) from 8.81 to 0.89 (*p* < 0.001)
Bayon-Calatayud et al., 2017 [36] Case study Pre-post test Level 5	Training with BCI-FES-VR system + 1 h of OT for the arm involved in the study.	5×/10 days	60 min	ND	Grip, strength, arm sensation, effort.	AIS, Borg scale, usability questionnaire, SCIM scale, EEG.	Grasp improved from 20 to 24 points in the affected arm and remained the same in the control arm. Arm strength and sensation did not change. SCIM scale improved from 28 to 42. Borg effort scale was 6.
Casadio et al., 2011 [37] Pre-Post test Level 4	4 cameras monitored the UL movements of people with SCI sitting in front of a monitor where they are asked to carry out specific movements.	2/3×/week	ND	GC: 9 sessions GI: 6–9 sessions	Reach, linear speed, rotational speed, force, ROM.	MMT, AIS, standard scale.	The MMT score improved significantly for all subjects (F (1,5) = 10; *p* = 0.02). The total isometric force exerted by the shoulder also improved for 5 of the 6 subjects with SCI.
King et al., 2013 [38] Pre-post Level 4	Participants used VR connected to a BMI to control their avatar to generate periods of walking and periods of idling.	1×/week	20 min	5 weeks	KMI (kinesthetic motor imagery) walking and inactive KMI, total time to complete course, number of successful stops.	AIS, EEG, FFT (Fast Fourier Transformation).	Online performance improved from 77.8 ± 13% to 85.7 ± 10.2%. Classification accuracy of idling and walking was estimated offline and ranged from 60.5% (*p* = 0.0176) to 92.3% (*p* = 1.36 × 10−20) across participants and days.
Leeb et al., 2007 [39] Case study Pre-post test Level 5	Immersive VR in which the patient practiced simulated driving moving the wheelchair along a street and stopping in front of avatars that engage in conversation if you get close enough to them.	ND	7 min	4 months	Motor Imaging (MI) of left and right hand and foot, time of each run, distance traveled, distance to the avatar, correct stops on avatars.	AIS, EEG (with a single threshold for MI and a resting threshold)	The subject was able to stop in 90% of the VR avatars of all his runs. In 4 runs, 100% performance was shown. Subject avatar distance: 1.81 m. Communication range 0.5 m to 2.5 m. It took 6.66 s. to move again after contact with an avatar.
Mason et al., 2004 [40] Controlled clinical trial Pre-post Level 4	Participants via VR played a video game in which they used a switch connected to the brain via EEG to turn the avatar to the left when the switch was turned on.	3×/week	60 min	2 weeks	Number of expected attempts with activation of a switch, number of attempts without activation of switch (TP, PT, NTP, NFP).	EEG (Electro-Cap), EOG (Electrooculography).	All 8 participants (4 with SCI) were able to control the switch. Switch activation rates ranged from 30 to 78%. FP between 0.5 and 2.2%. Changes were not significant.
Nicolelis et al., 2022 [41] Randomized controlled clinical trial Pre-post Level 1	Participants are neurologically matched using noninvasive BCI assisted locomotion, VR and tactile feedback.	2×/week	45 min	13/14 weeks	Proprioception, vibration perception, spinal cord status, sitting and standing avatar gait performance.	AIS, EEG (16 channels), BMI, Open Vibe, MRI, *T*-test, Pinprick, one-way ANOVA.	A higher delta score was observed for the L + B group compared to the LOC group for the Pinprick test. 3 of the 4 L + B participants, at the end of the protocol were classified as AIS C. One participant in the LOC group went from AIS A to AIS C. Accuracy was on average 72% higher *p* < 0.054 Improvements in P4, P6 and P7 performance.
Pais-Vieira et al., 2022 [42] Case Study Pre-post Level 5	Patient connected to a BCI enters a VR equipped with glasses, tactile and thermal feedback sleeves, headphones, and controls where the patient must relate shapes with colors to the thought of walking or not walking. Choosing the scenario where the patient wants to be during the VR.	2×/week	70–90 min	5 weeks	Comfort with thermal-tactile sleeves, pain, sensations at home, sensitivity, perception of body qualities, volitional control of movements, tactile perceptions.	AIS, Headset with headphones, two hand controllers, two thermal-tactile sleeves and tactile stimulation patterns for the arm, EEG (16 channels), Open Vibe, Faces pain scale, verbal pain intensity scale, VAS pain scale, SSQ.	Session performance started at 80% and peaked at 100% in session #6. The average VAS pain scale was 6.29, the faces pain scale was 5.21, and verbal pain was scored as moderate 6/7. Performance of sessions without cuff *p* = 0.2857. Differences between sessions with and without thermal-tactile sleeve in terms of pain Faces pain scale *p* = 0.3379 and VAS scale *p* = 0.1632.
Salisbury et al., 2016 [43] Pre-post Level 4	The basic game consists of the participants being able to move cubes by means of the BCI while entering a state of neutral condition.	ND	ND	ND	Cognitive functioning, intelligence, mood, mood, physical state, pain, disability, perceived pain, ability to avoid and focus on activity, EEG.	AIS, Wechsler Scale, oral traces test, Wechsler Reading, PHQ-9, McGill questionnaire, Tellegen Absorption Scale, EEG.	The participants successfully completed the game and showed enjoyment of the experience, on a scale of 1 to 100. The average enjoyment was 79.2. The study showed feasibility, although there were failures in the technology used. Number of successful trials in McGill questionnaire (*p* < 0.001). Mean power level achieved in all tests with the McGill questionnaire (*p* = 0.009).
Tidoni et al., 2016 [11] Post- test Level 4	CG and IG: immersive VR of mathematical game with board and proprioceptive stimulator on the biceps brachii tendon with video feedback recorded by robot.	12×/ND	6 min	ND	Results of user experience questionnaire, optimization calls and data transfer rate.	AIS UE OC ITR EEG.	Patient 1: lower task accuracy than CG and higher OC and lower RTI (*p* < 0.022). Patient 2: only VR. UE, OC and ITR did not differ from CG. Patient 3: did not differ from CG in the robot scenario, although UE and ITR scored lower in VR.
Wang et al., 2012 [44] Post-test Level 4	Participants entered a VR environment featuring a flat grassland where there were 10 NPCs in a straight line. Subjects used KMI to move forward and idle to stop next to each NPC (third person view).	ND	10 min	ND	Completion time, successful stops.	AIS, EEG (63 channels), EMG.	Average off-line training performance among the subjects was 77.2 ± 11.0%, with a range of 64.3% to 94.5%. subjects were 77.2 ± 11.0%, with a range of 64.3% to 94.5%. Average online performance was 85% successful stops and 303 s. completion time (ideal is 211 s). All subjects achieved performances that were significantly different from random walking (*p* < 0.05) in 44 of the 45 online sessions.

AIS: American Spinal Injury Association Impairment Scale; ANOVA: Analysis of Variance; BCI: Brain–computer Interface; BI: Barthel Index; BMI: Body Mass Index; CBMI: Customized body–machine interface; CG: Control Group; CT: Conventional therapy; EEG: Electroencephalogram; EMG: Electromyogram; FES: Functional electrical stimulation; FFT: Fast Fourier Transform; FP: False Positive; IG: Intervention Group; ITR: Information transfer rate; KMI: Kinesthetic motor imagery; L + B: Neurorehabilitation protocol integrating assisted locomotion with a noninvasive brain–machine interface; LOC: Intensive assisted locomotion training; MI: Motor Imagery; MMT: Modified manual muscle test; MRI: Magnetic Resonance Imaging; ND: Not described; NFP: Number of switch activations when the user was in a no control state; NPC: Non player character; NTP: Number of intended attempts with a switch activation; OC: Optimization Calls; OT: Occupational therapy; P: Participant; PHQ-9: Patient Health Questionnaire-9; PT: Physical Therapy; ROM: Range of motion of joints; SCI: Spinal cord injury; SCIM: Spinal cord injury independence measure (0–100); SSQ: Simulator Sickness Questionnaire; TP: True positive rate; UE: User experience questionnaire; UL: Upper limbs; VAS: VAS-like pain scale/Upper airways; VR: Virtual Reality.

## Data Availability

The datasets analyzed for this study can be found in the manuscript and Appendix A.

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
