# Peer review of "Effectiveness of the Combined Use of a Brain–Machine Interface System and Virtual Reality as a Therapeutic Approach in Patients with Spinal Cord Injury: A Systematic Review"

_healthcare, 2023, doi:10.3390/healthcare11243189_

Round 1

Reviewer 1 Report

Comments and Suggestions for Authors

In the manuscript “Effectiveness of the combined use of a brain-machine interface system and virtual reality as a therapeutic approach in patients with spinal cord injury: a systematic review “ the authors have presented some basic findings. The study is poorly designed and is only a descriptive survey. There is not novelty in this study.

Comments on the Quality of English Language

In the manuscript “Effectiveness of the combined use of a brain-machine interface system and virtual reality as a therapeutic approach in patients with spinal cord injury: a systematic review “ the authors have presented some basic findings. The study is poorly designed and is only a descriptive survey. There is not novelty in this study.

Reviewer 2 Report

Comments and Suggestions for Authors

Dear Authors, 

I am grateful for reviewing this detailed manuscript. 

The paper comprises a systematic review analyzing the association of brain-machine interface and virtual reality in patients with spinal cord injury. 

The work has a robust methodology and is well written. The images (tables and schemes) are appropriate. Supplementary materials are useful.

I cannot identify any critical point. 

I leave here some minor suggestions:

- Line 267: please, CG and IG should be written in extenso when they first appear

- Line 352: I would remove "regarding the limitations" because it is a noisy repeat

Thank you for considering my review report

Best Regards

Comments on the Quality of English Language

Minor english editing

Reviewer 3 Report

Comments and Suggestions for Authors

This systematic review explores the effectiveness of combining a brain-machine interface (BMI) system with virtual reality (VR) as a therapeutic approach for individuals with spinal cord injuries (SCI). The study, conducted through a comprehensive literature review, identifies eleven relevant articles from a pool of 82, primarily using electroencephalography (EEG) to measure outcomes. The combined use of BMI and VR shows statistically significant improvements, particularly in upper limb function, including enhanced shoulder and upper arm mobility and strengthened muscles. While the results are promising, the authors emphasize the need for further research with larger sample sizes, long-term follow-ups, and a focus on categorizing patients based on the severity of SCI to tailor treatment protocols effectively. The conclusion underscores the potential of BMI-VR therapy for SCI patients but advocates for continued investigation and refinement of protocols for optimal outcomes.
While the manuscript is well-constructed, there are some minor points that require attention: the initial mention of EEG is not  abbreviated in the abstract or text, ADL need not be abbreviated as it appears only once, and a careful revision of abbreviations, such as SCI and ASIA, is recommended. Additionally, the authors are advised to refrain from providing an extensive explanation, constituting a whole paragraph, of ASIA assessment principles within the introduction. Once these minor points are addressed, the manuscript can be accepted for publication.

Comments on the Quality of English Language

none

Reviewer 4 Report

Comments and Suggestions for Authors

The authors presented a systematic review aimed at evaluating the effectiveness of the combined use of virtual reality and the brain-machine interface in the treatment of spinal cord injuries. Articles were selected based on strict inclusion and exclusion criteria. Eleven articles were selected and the authors concluded that most of the articles analysed used the EEG as a measurement instrument for the assessment of various parameters, and most studies have shown improvements.

The authors diligently carried out a very interesting project, contributing a lot to the assessment of the treatment of patients after partial spinal cord injuries.

I will advise on the minor improvements or corrections.

In the Abstract section, more data are necessary in lines 21-24 on the methods of the analysis of the results (please list,), although in the M&M  section they are rich in principles. The conclusion in lines 26-27  (...Nonetheless, further research is needed with larger sample size and long-term follow-up...) is very general; maybe such studies will be performed, maybe not.

When discussing the state of the art in the Introduction chapter in lines (62-64), methods of assessing from clinical neurophysiology and their findings (except EEG) in patients with partial spinal cord injuries have been described very generally. Some refs. could be cordially invited.

I am not convinced that the AI actually accurately retrieved all the works relating to the problem. The authors should discuss this possibility more fully in the study limitation. Most of the works analyzed in detail concern the treatment of neuropathic pain resulting from spinal injuries. What about sensory and motor dysfunctions? Surprisingly, no studies were detected (analyzed) estimating the effectiveness of treatment using somatosensory and motor evoked potential tests. Perhaps it comes from the search narrowing described in 2.2. Selection Criteria. The authors write a lot about the ASIA scale in the Introduction, but little about assessing the conduction of sensory and motor nerve impulses.
